# Genomic surveillance of SARS-CoV-2 in Puerto Rico enabled early detection and tracking of variants

Gilberto A. Santiago [1✉], Betzabel Flores[1], Glenda L. González[1], Keyla N. Charriez[1], Limari Cora Huertas[2], Hannah R. Volkman [1], Steven M. Van Belleghem[2], Vanessa Rivera-Amill[3], Laura E. Adams[1], Melissa Marzán[4], Lorena Hernández[4], Iris Cardona[4], Eduardo O'Neill[5], Gabriela Paz-Bailey[1], Riccardo Papa[2] & Jorge L. Muñoz-Jordan[1]

## Abstract

**Background** Puerto Rico has experienced the full impact of the COVID-19 pandemic. Since SARS-CoV-2, the virus that causes COVID-19, was first detected on the island in March of 2020, it spread rapidly though the island's population and became a critical threat to public health.

**Methods** We conducted a genomic surveillance study through a partnership with health agencies and academic institutions to understand the emergence and molecular epidemiology of the virus on the island. We sampled COVID-19 cases monthly over 19 months and sequenced a total of 753 SARS-CoV-2 genomes between March 2020 and September 2021 to reconstruct the local epidemic in a regional context using phylogenetic inference.

**Results** Our analyses reveal that multiple importation events propelled the emergence and spread of the virus throughout the study period, including the introduction and spread of most SARS-CoV-2 variants detected world-wide. Lineage turnover cycles through various phases of the local epidemic were observed, where the predominant lineage was replaced by the next competing lineage or variant after ~4 months of circulation locally. We also identified the emergence of lineage B.1.588, an autochthonous lineage that predominated in Puerto Rico from September to December 2020 and subsequently spread to the United States.

**Conclusions** The results of this collaborative approach highlight the importance of timely collection and analysis of SARS-CoV-2 genomic surveillance data to inform public health responses.

### Plain language summary

The COVID-19 pandemic reached Puerto Rico in March 2020. To understand the impact of SARS-CoV-2 on Puerto Rico, we formed a partnership with universities and local government to study the genetic sequence of viruses sampled from infected people between March 2020 and September 2021. Our results show that the local epidemic was initiated and sustained by frequent importation of a wide diversity of SARS-CoV-2 lineages and variants, some of which circulated for some time in the island. We also detected a lineage of SARS-CoV-2, named B.1.588, that was first detected in Puerto Rico and subsequently spread to the United States. This study highlights the importance of the study of viral genetic data to inform public health responses.

[1] Centers for Disease Control and Prevention, National Centers for Emerging and Zoonotic Infectious Diseases, Division of Vector Borne Diseases, Dengue Branch, San Juan, Puerto Rico. [2] University of Puerto Rico—Río Piedras, Department of Biology, Molecular Sciences and Research Center, San Juan, Puerto Rico. [3] Ponce Health Sciences University, Ponce Research Institute, Department of Basic Sciences, Ponce, Puerto Rico. [4] Puerto Rico Department of Health, Epidemiology Office, San Juan, Puerto Rico. [5] Centers for Disease Control and Prevention, Office of Island Affairs, Center for State, Tribal, Local, and Territorial Support, Atlanta, GA, USA. ✉email: GSantiago@cdc.gov

The current coronavirus disease 2019 (COVID-19) pandemic, caused by the severe acute respiratory syndrome coronavirus 2 (SARS-CoV-2), was initially declared a Public Health Emergency of International Concern in January 2020[1,2]. Despite global efforts to interrupt transmission chains with quarantine, isolation, and travel restrictions at the onset of the pandemic, SARS-CoV-2 spread rapidly across the globe, creating a global pandemic and threat to human health worldwide. By November 15th, 2021, the World Health Organization (WHO) reported 253 million confirmed COVID-19 cases in 222 countries and over 5 million deaths[3,4]. SARS-CoV-2 reached all 50 states of the United States and associated territories, including Puerto Rico, by March 2020, after multiple introductions by travelers with infection[5–7]. The rapid spread across the United States was primarily propelled by interstate transmission chains and air travel to the associated territories[6,8,9].

SARS-CoV-2 is an enveloped virus with a single-stranded positive-sense RNA genome of ~30,000 base pairs. During replication, a virus-encoded exonuclease provides a proof-reading activity that contributes to the observed low mutation rate and stable genome[10,11]. Nevertheless, the unprecedented spread of SARS-CoV-2 globally and the wealth of genomic sequence data available through the international initiative for genomic studies and surveillance has facilitated phylodynamic approaches to infer viral evolutionary rate, growth rate, and the estimated time of origin for specific outbreaks[11]. Studies have revealed that the viral genome has been accumulating mutations of concern, especially in the spike protein region, which confer phenotypes with increased fitness and pathogenicity[12–14]. Increased infectivity, resistance to monoclonal antibody therapy and evasion of the immune response were among the most frequently observed phenotypes attributed to WHO-monitored variants; these phenotypes often dominated transmission and replacement of other lineages upon emergence[15–18]. The Variant Being Monitored (VBM) B.1.1.7 (Alpha) was first identified in the United States in late December 2020 and was then characterized by a considerable increase in COVID-19 incidence associated with increased infectivity and occasionally more severe disease manifestations that increased hospitalization rates[19,20]. Alpha became the dominant variant, especially in Europe and the United States, until the emergence of Variant of Concern (VOC) B.1.617.2/AY.x (Delta), first identified in the United States in May 2021, which developed into a prominent variant with an apparent higher virulence and pathogenic phenotype[21–23]. Because of the potential for increased transmissibility, morbidity mortality, and decreased efficacy of vaccines and other intervention strategies, monitoring the spread of variants (VBMs and VOCs) rapidly became a public health concern and priority[24,25].

Puerto Rico, an unincorporated territory of the United States, is a densely populated island and a popular tourist destination located in the Caribbean basin. SARS-CoV-2 was first identified in Puerto Rico on March 13th, 2020, in two European travelers who arrived on a cruise ship and in one local resident who had close contact with family members with recent travel history. Additional travel-related and local cases were confirmed within the following weeks[26]. In response to the emerging threat, the government of Puerto Rico executed the most restrictive (compared to the United States) national stay-at-home order on March 15th, 2020, to mitigate transmission while preparing the public health infrastructure for the imminent impact[27,28]. Travel restrictions imposed by the United States during the initial pandemic minimized international traffic to Puerto Rico, although domestic travel from the United States continued. Puerto Rico represents a unique epidemiologic setting in a geographically isolated location (an island), but with a regular influx of travelers mostly from the United States. This is an ideal setting to monitor introduction and spread of SARS-CoV-2 variants and answer questions to help inform SARS-CoV-2 spread and disease prevention strategies. Puerto Rico's public health response incorporated extensive molecular surveillance to the increased laboratory capacity, which presented a unique opportunity to study the impact of SARS-CoV-2 variant turnover, local dissemination, and evolution during a period of changing epidemiology and public health responses.

In response to the impending local epidemic, we established a partnership with the local health authorities and academia to conduct a genomic surveillance initiative to sample complete genomes of SARS-CoV-2 across the island through time, monitor lineage circulation, and understand the genomic epidemiology of the COVID-19 pandemic in Puerto Rico. This report presents the results from 19 months of genomic surveillance and phylogenetic analyses, which identified multiple introduction events that propelled the rapid expansion and persistent transmission of the virus on the island and lead to the establishment of an autochthonous lineage between August 2020 and January 2021.

## Methods

**Epidemiological data.** We retrieved the number COVID-19 cases reported by the Puerto Rico Department of Health (PRDH) from March 2020 to 30 September 2021 from the PRDH database dashboard on 1 December 2021 available here https://covid19datos.salud.gov.pr/. The collection of cases includes cases classified as confirmed (by molecular tests) or probable (by antigen tests) and plotted by date of sample collection.

**Patient sample selection.** Nasopharyngeal swab samples preselected for genomic surveillance were received from COVID-19 passive surveillance conducted by PRDH, the Ponce Health Sciences University (PHSU), and hospital-based acute febrile illness surveillance conducted by the Centers for Disease Control and Prevention (CDC) Dengue Branch. A total of 785 samples were collected from March 2020 to September 30th, 2020, from the seven health regions of the island, including 63 out of the 78 municipalities, and selection criteria included all samples with SARS-CoV-2 detected by reverse-transcriptase polymerase chain reaction (RT-PCR), viral load (CT < 28) and sufficient residual sample volume stored at −80 °C[29]. All samples were de-linked from patient identifiable information and processed under the guidelines approved by the CDC and Ponce School of Medicine institutional review boards (IRB) protocol 6731, which waived the need for informed consent for sequencing of residual samples.

**Lineage frequency analysis.** The frequency of SARS-CoV-2 lineage detection in Puerto Rico was calculated using the total number of SARS-CoV-2 genomes published in the Global Initiative on Sharing All Influenza Data (GISAID) (https://www.gisaid.org) with collection dates ranging between March 1st, 2020 and September 30th, 2021. All complete genome sequences and metadata were retrieved from the GISAID database as of October 31st, 2021. The dataset was filtered for complete genome data, high-coverage data, and complete collection date for a final dataset of 2514 entries. Lineage assignment on GISAID was determined by the Phylogenetic Assignment of Named Global Outbreak Lineages (Pangolin)[30,31]. R with ggplot package was used to calculate lineage frequency and plot the graph focusing on the following lineages of interest: B.1.1.7 (Alpha), P.1 + P.1.1 (Gamma), B.1.588, Delta (B.1.617.2+AY.x), B.1.427 + B.1.429 (Epsilon), B.1.526 (Iota), B.1.621/1 (Mu), and all other Pangolin-designated B lineages grouped as Other. No genomes collected in May 2020 have been published in GISAID by October 31st, 2021.

**Complete genome sequencing and assembly**. Complete SARS-CoV-2 genome sequences were generated directly from clinical nasopharyngeal samples. Viral RNA was extracted from viral transport media using the automated MagNA Pure 96 system (Roche) with the MagNA Pure 96 DNA and Viral Nucleic Acid Small Volume Kit (Roche) following manufacturer-recommended protocols for 0.2 mL sample input volume and 0.1 mL RNA elution volume. MP96 external lysis buffer was used to pre-treat the samples for neutralization and assist the lysis process. First strand cDNA was synthesized with random hexamers using SuperScript IV reverse transcriptase (ThermoFisher), and tiling PCR amplicons were generated using Q5® high-fidelity DNA polymerase (New England Biolabs) and the ARTIC nCoV-2019 V3 primer scheme purchased from Integrated DNA Technologies (https://github.com/artic-network/artic-ncov2019/blob/master/primer_scheme/nCoV-2019/V3/nCov-2019.tsv). Candidate samples for sequencing presented clearly visible bands of target size (~400 bp) in DNA gel electrophoresis for both primer pools. PCR products were purified with AMPure XP magnetic beads (Beckman Coulter) and quantified using Qubit 4.0 fluorometer (ThermoFisher). DNA libraries were generated using the NEBNext Ultra II DNA Library Prep Kit for Illumina (New England Biolabs), reducing all reagents volumes to 25% from the manufacturer's recommended protocol to increase throughput. The resulting products were screened for size and quality using the Bioanalyzer 2100 instrument (Agilent Technologies) and quantified with Qubit 4 fluorometer (ThermoFisher). Qualifying libraries were pooled and run in the MiSeq sequencer instrument (Illumina) using the MiSeq Reagent Kit v3 in 600-cycle program.

The resulting sequence reads were screened for quality, trimmed, and assembled into complete consensus SARS-CoV-2 genomes using the Genome Detective Virus Tool v1.136[32] (https://www.genomedetective.com) and assembly confirmed with iVar[33]. The Pangolin COVID-19 Lineage Assigner tool was used for lineage assignment[34] (https://pangolin.cog-uk.io). A total of 753 samples were sequenced with more than 95% genome coverage at a minimum of 10x sequence depth. All sequence data obtained for this study was submitted to GISAID, accession numbers available in Supplementary Data 1.

**Phylogenetic analysis**. Our Puerto Rico SARS-CoV-2 genomes dataset was analyzed against a diverse panel of genomes from across the world which provide regional phylogenetic context. Initially, we downloaded the Genomic Epidemiology metadata package for all entries from GISAID on August 18th, 2021 to screen genomes for subsampling. However, due to the large number of genomes available in GISAID, we downloaded and combined the following pre-sampled datasets for regional studies: NextRegion-North America, NextRegion-South America, and NextRegion-Global. We then used the standard ncov augur/auspice multiple input workflow available in the Nextstrain platform[35] (https://github.com/nextstrain/ncov) to subsample contextual genomes and phylogenetic inference with time-stamped trees. The custom subsampling scheme program selected 2611 contextual genomes from the United States, North America, the Caribbean, Central America, South America, Africa, Europe, Asia, and Oceania, with higher proportions from The Americas and selected based on collection dates and genetic proximity to our Puerto Rico dataset. The combined dataset of 3364 genomes was aligned using MAFFT[36] and a global maximum likelihood (ML) phylogenetic inference was reconstructed with IQ-TREE[37]. The ncov workflow then transferred the ML tree to TreeTime[38] for time calibration and ancestral state reconstruction of the tree topology at constate rate of $8 \times 10^{-4}$ nucleotide substitutions per site per year. The resulting global ML tree was visualized with Nextstrain auspice[35] and annotated with

iTol for region of origin and emerging variants[39]. Subsampling from the Genomic Epidemiology metadata package retrieved in August and from the combined NextRegions produced phylogenetic inference trees with similar topologies. A list of all the sequences used in this study, including sequence labels and authors can be found in the "Data availability" section.

Selected lineages of interest were studied further by reconstruction of phylogenetic focus trees. For the B.1.588 lineage-focused tree, we selected all B.1.588 genomes published in GISAID by October 31st, 2021. Contextual B.1 lineage genomes were selected based on phylogenetic clustering near the base of the B.1.588 clade in the global tree and by temporal proximity to the date range of B.1.588 circulation between June 2020 and January 2021. Maximum likelihood phylogenetic trees were reconstructed with the resulting dataset of 239 genomes under the GTR + G + I nucleotide substitution model and 1000 bootstrap replicates using IQ-TREE v1.6.12[37]. The resulting tree topology and node support were compared to Bayesian maximum clade credibility (MCC) tree reconstruction using BEAST v1.10.4[40]. Briefly, we used time-calibrated genomes trimmed to coding sequence with sample collection dates and the nucleotide substitution model parametrized using Yang96 model under strict molecular clock, and Bayesian Skyline coalescent model. Markov chains were run for a total of 100 million steps with sampling every 10,000 steps in the chain. Run results were evaluated in Tracer (http://tree.bio.ed.ac.uk/software/tracer/) to ensure stationary parameters with statistical errors reflected in 95% highest probability density ranges with effective sample size (ESS) higher than 200 for each tree prior. MCC trees were generated in TreeAnnotator from BEAST package after discarding 10% of runs as burn-in. The resulting ML and MCC trees were visualized in FigTree v1.4.4 (http://tree.bio.ed.ac.uk/software/figtree).

For the B.1.1.7 (Alpha VBM) lineage-focused tree, we selected all B.1.1.7 + Q.x designated genomes from the global tree and supplemented the dataset with additional B.1.1.7 + Q.x genomes from Puerto Rico and the United States to understand the lineage emergence and spread in the island. A custom subsampling scheme was selected on the Nextstrain ncov workflow to select genomes from samples collected between November 1st, 2020 and February 28th, 2021. Maximum likelihood phylogenetic trees were reconstructed with the resulting dataset of 729 genomes under the GTR + G + I nucleotide substitution model and 1000 bootstrap replicates using IQ-TREE v1.6.12[37]. The resulting ML tree was visualized in FigTree v1.4.4 (http://tree.bio.ed.ac.uk/software/figtree).

For the B.1.617.2 (Delta VOC) lineage-focused tree, we selected all B.1.617.2 + AY.x designated genomes from the global tree and supplemented the dataset with additional B.1.617.2 + AY.x genomes from Puerto Rico to understand lineage spread and sublineage clustering patterns across the island. The genome selector Python script designed by Anderson Brito (https://github.com/andersonbrito/ncov) was used to select additional Delta-designated genomes from Puerto Rico with collection dates range from June 1st, 2021 to September 30th, 2021. Maximum likelihood phylogenetic trees were reconstructed with the resulting dataset of 815 genomes under the GTR + G + I nucleotide substitution model and 1000 bootstrap replicates using IQ-TREE v1.6.12[37]. The resulting ML tree was visualized in FigTree v1.4.4 (http://tree.bio.ed.ac.uk/software/figtree).

The date of the most recent common ancestor (tMRCA) determined by the Nextstrain ML phylogenetic inference was confirmed with Bayesian coalescent analyses for B.1.588, Alpha VBM and Delta VOC lineage trees. Due to the large size of the datasets, each focus tree was reduced by tree-pruning to datasets with <150 genomes. tMRCA analyses were performed with BEAST v1.10.4 under a strict molecular clock fixed at $8 \times 10^{-4}$ substitutions per site per year and 150 million Markov chains sampling every

10,000 steps. Results were evaluated in Tracer (http://tree.bio.ed.ac.uk/software/tracer/) for convergence and ESS >200.

**Reporting summary**. Further information on research design is available in the Nature Research Reporting Summary linked to this article.

## Results

**Local epidemic and variant detection**. During March–July 2020, the number of confirmed COVID-19 cases remained low, associated with the strict stay-at-home order. After the order was lifted in June 2020, the number of cases increased steeply during the summer of 2020, initiating the local epidemic (Fig. 1a). Since then, we observed three epidemic waves with high points in November 2020, April 2021, and August 2021. By September 30th, 2021, the Puerto Rico Department of Health and the Centers for Disease Control and Prevention reported over 181,599 confirmed cases[26].

We conducted genomic surveillance for 19 months since March 2020, where, each month, we sequenced SARS-CoV-2 positive specimens from recently symptomatic and asymptomatic patients residing in 63 of the 78 municipalities, covering all seven health regions of the island. The frequency of lineages detected was calculated periodically from all viral genomic sequences from Puerto Rico published in GISAID, including our sequence datasets, and reported to the PRDH to inform case investigations and surveillance (Fig. 1b). Our data comprise the genomes from the initial SARS-CoV-2 confirmed infections detected in March 2020, which included three European tourists that arrived on the island in a cruise ship and eight residents with no recent travel history declared. PANGO lineage assignment after sequencing identified lineage A.1, a lineage predominant in Europe at the time, in the three travelers with infection, while lineages B.1 and B.1.1 were identified in residents with infection. B.1x lineages predominated in the United States and the Americas. The initial phase of the epidemic was characterized by the detection of a wide diversity of B.1x lineages that circulated at low frequency for short periods of time, suggesting that the local epidemic was initiated by multiple introduction events. In August 2020, we detected the emergence of lineage B.1.588 in various municipalities of the island. Lineage B.1.588 rapidly became the predominant lineage in Puerto Rico, circulating at high frequency for ~4 months and causing the first epidemic wave in November 2020 (Fig. 1a, b). Circulation of lineage B.1.588 declined during the first wave of the epidemic in the winter of 2020, a season of local holiday festivities and frequent travel. During this season, the diversity of B.1x lineages increased, and the first emergent variants were detected in the island, VBM B.1.427/429 (Epsilon) in December 2020 and Alpha in January 2021 concordant to variant emergence in the United States (Fig. 1b). Concurrently, the first stage of the COVID-19 vaccination campaign in Puerto Rico started in mid-December 2020 for the elderly population and first responders. A steep reduction in confirmed cases was observed in the following months despite the introduction of VBMs B.1.526 (Iota) and P.1/1.1 (Gamma) in February and March 2021 respectively and the predominant circulation of Alpha in March 2021 (Fig. 1b). The second wave of the epidemic was observed in April 2021 with Alpha predominating (Fig. 1a, b). Though other emerging variants continued to be detected, the frequency of detection remained low, and Alpha predominated for ~3–4 months. The second stage of the COVID-19 vaccination campaign started in April 2021 for all adults and was immediately followed by a sharp decrease in confirmed cases, a period in which ~50% of the population had received at least one dose of the vaccine[26] (Fig. 1a). VOC B.1.617.2/AY.x (Delta) was first detected in June 2021, concordant with the emergence in the

United States, and rapidly dominated transmission. During the same period, we detected the emergence of VBM B.1.621 (Mu), which caused a small local outbreak in the western part of the island, as well as a modest increase in Gamma infections (Fig. 1b). The third epidemic wave was observed in August 2021, coinciding with a summer of increased travel and the removal of local government-imposed restrictions on business indoor occupant capacity and public gatherings (Fig. 1a, b). During this period, most COVID-19 cases in Puerto Rico were caused by Delta and ~18 Delta sub-lineages were detected in the island, with AY.3 as the most frequently sampled sub-lineage (Fig. 1c). These data provide further evidence of the multiple importations received during this period of the epidemic. By September 30th, 2021, a steep decrease in confirmed cases was observed, a point in which more than 77% of the population had received at least one dose of the vaccine (Fig. 1a).

**Phylogenetic reconstruction of the local pandemic**. This study generated 753 complete genomes from viruses sampled between March 2020 and September 2021. Our dataset was combined with 2611 publicly available genomes in GISAID to understand the emergence and spread of the viruses circulating in Puerto Rico in a global context. We reconstructed the local and regional epidemic using a time-calibrated phylogenetic tree inferred with maximum likelihood (Fig. 2). This global phylogenetic analysis estimated that the initial SARS-CoV-2 introductions occurred between February 19 and March 16, 2020. Most viral genomes from Puerto Rico descend or are closely related to genomes from the United States. However, we were unable to determine the precise origin at the state level due to the limited sampling during the emergence period and subsequent low circulation. The resulting tree topology inferred viral sequences from Puerto Rico scattered across the global tree, smaller short-lived monophyletic clusters, and larger monophyletic clusters that suggest sustained transmission of a particular genotype. Our analysis also showed the emergence and evolution of the SARS-CoV-2 variants detected in Puerto Rico. Multiple monophyletic clusters of Puerto Rican sequences were inferred within the clades formed by each emergent variant and the size of the clades is proportional to the frequency of genomes sampled in the island (Figs. 1b and 2). The observed clustering patterns in the phylogenetic trees and the rapid increase in frequency following initial detection indicate multiple virus introductions with swift expansion across the island in a short period of time.

**Detection and spread of autochthonous lineage B.1.588**. During the initial phase of epidemic transmission, we detected the emergence of an autochthonous lineage, B.1.588, which rapidly spread across the island. Based on GISAID data and cov-lineages.org reports (https://cov-lineages.org/lineages), lineage B.1.588 was first detected in Puerto Rico on August 2nd, 2020: sequence EPI_ISL_1168693. Initially, lineage B.1.588 circulated only in Puerto Rico, accounting for approximately half of the viruses sampled in the island in September 2020. B.1.588 quickly became the predominant lineage in Puerto Rico during the first epidemic wave, circulating for 4 months until it was replaced by the emergence of Alpha in January 2021 (Fig. 1b). This study sequenced 97 out of the 115 B.1.588 genomes from Puerto Rico found in GISAID. To understand the emergence and spread of this lineage, we reconstructed a focused phylogenetic tree using maximum likelihood and Bayesian inference with 103 B.1.588 sequences from Puerto Rico, 58 B.1.588 sequences from the United States and an additional set of 77 B.1 lineage sequences closely related to B.1.588 (Fig. 3). Our analysis estimated that lineage B.1.588 diverged from its parental lineage B.1 between May 21st, 2020, and July 16th, 2020 in Puerto Rico, after

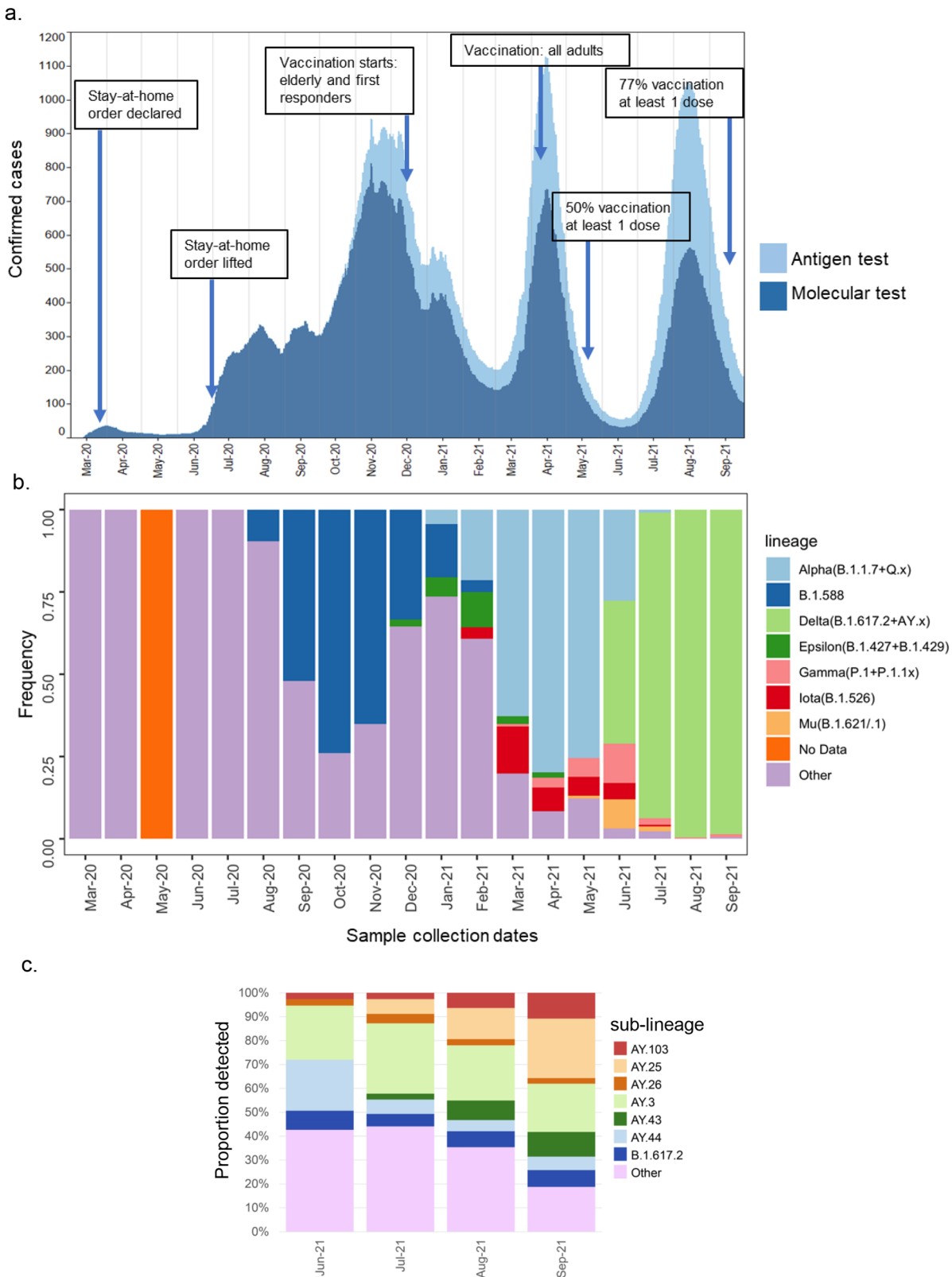

the appearance of two non-conservative mutations: T20I in the spike protein and M234I in the nucleocapsid protein. Subsequently, lineage B.1.588 spread broadly to the United States, mainly in New York, Texas, Florida, and California, where it circulated until May 2021 concomitant with a diversity of other lineages and variants. More than 990 B.1.588 genomes have been reported in the United States.

**Emergence of SARS-CoV-2 variants**. VBM Alpha was first detected in Puerto Rico in January 2021, co-circulating with local predominant lineage B.1.588 and other B lineages at a lower rate (Fig. 1b). Notably, this VBM replaced the well-established autochthonous lineage B.1.588. The emergence and epidemiology of Alpha in Puerto Rico resembled the patterns observed in the United States, with rapid spread and a sharp increase in confirmed

**Fig. 1 Epidemiology and dynamics of SARS-CoV-2 lineage turn-over in Puerto Rico. a** Graphical representation of the number of daily SARS-CoV-2 cases confirmed by antigen tests (light blue) and molecular tests (dark blue) reported by the PRDH from March 2020 to September 30th, 2021, shown as a 21-day rolling average. Arrows indicate the timeline of government responses and vaccination milestones. **b** Proportion of all SARS-CoV-2 lineages and emerging variants detected in Puerto Rico and published in GISAID from March 2020 to September 30th, 2021 (n = 2514 sequences after filtering for high-coverage genomes with complete sampling dates). Non-VBM/VOC lineages (n > 63) are categorized as a collective labeled "Other", except for local lineage B.1.588 due to high frequency and focus of this study. No genomes were obtained during May 2020. **c** Proportion of all Delta sub-lineages detected in Puerto Rico published in GISAID until September 30th, 2021 (n = 1360 genomes). Sub-lineages with more than 5% detection are represented by individual color, whereas sub-lineages with <5% detection are categorized as a collective labeled "Other".

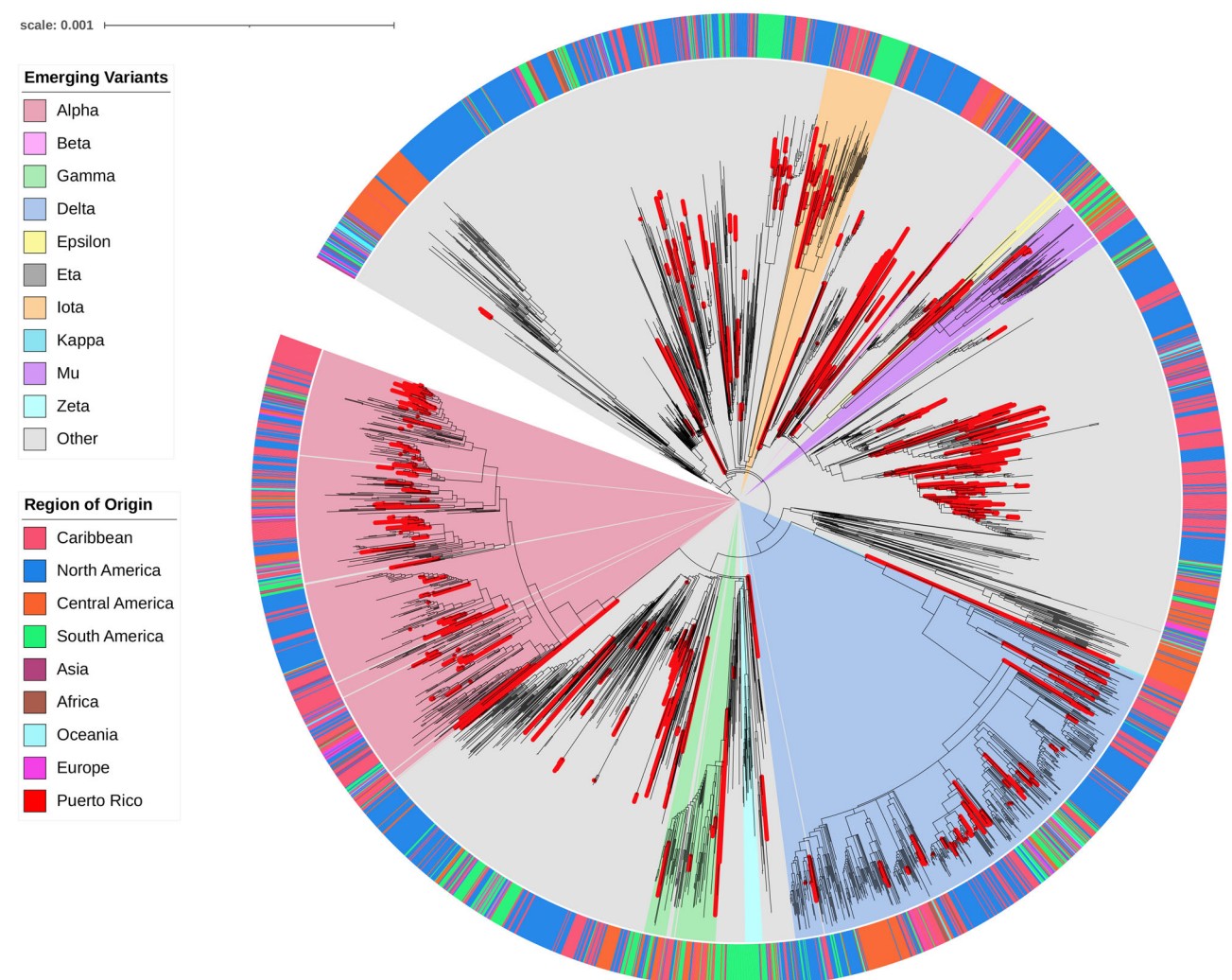

**Fig. 2 Phylogenetic reconstruction of local SARS-CoV-2 epidemic in Puerto Rico in a global context.** Maximum likelihood tree inferred with 3364 complete genomes including 753 viral genomes from Puerto Rico sampled between March 23rd, 2020 and September 30th, 2021 (red branches) combined with 2611 complete genomes retrieved from GISAID during the same period to provide a global backdrop with a higher focus on the Americas region. Node structure is supported by 1000 bootstrap replicates. Branches marked in red represent taxa from Puerto Rico. The outer ring is color-coded by region of origin. The inner wedges are color-coded to represent emerging variants of interest or concern. The phylogenetic tree is rooted in Wuhan/WH01/2019 and Wuhan/Hu-1/2019 reference genomes.

cases[19,20,41]. To understand the emergence and spread of Alpha in Puerto Rico, we inferred a maximum likelihood phylogenetic-focused tree with all Alpha genomes obtained in our dataset in addition to a subset of other Alpha genomes from Puerto Rico, the United States, and a regional context backdrop (Fig. 4). The resulting inference estimated that the emergence of Alpha in Puerto Rico may have occurred between November 6th, 2020, and December 31st, 2020. Tree topology showed multiple monophyletic clusters of Puerto Rican sequences diverging across a period of 4–5 months of circulation. The larger clusters of Puerto Rican

sequences suggest that local transmission of specific Alpha geno-types was sustained, succeeding after multiple introduction events. Most of these clusters were associated with sequences from the United States, suggesting that multiple introductions occurred over a period of 5–6 months, propelling the local transmission of this variant. We also found a subset of Puerto Rican sequences associated with sequences from the Caribbean and the Americas but low node support impaired resolution of transmission patterns.

VOC Delta was first detected in Puerto Rico in June 2021, during a period when SARS-CoV-2 transmission was declining,

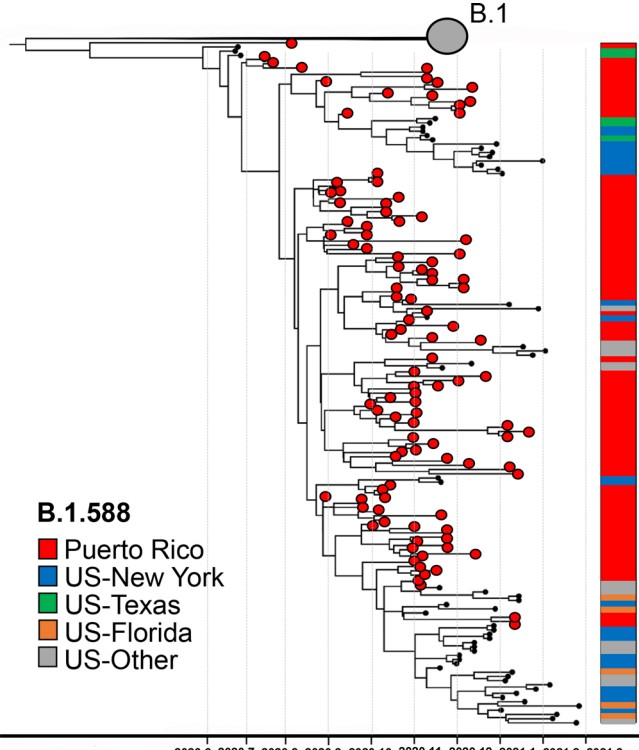

**Fig. 3 Emergence of autochthonous lineage B.1.588.** Phylogenetic reconstruction of monophyletic lineage B.1.588 using Bayesian maximum clade credibility tree inferred with 239 complete genomes including 130 genomes from Puerto Rico (103 B.1.588 genomes) sampled between July 2020 and March 2021. Node support was tested by posterior probability. The gray circle represents B.1 viral genomes from Puerto Rico and the United States that cluster basal to the B.1.588 monophyletic lineage. Red circle taxa tips represent viral genomes from Puerto Rico. Colored shade bar on the right of the tree indicates the taxa region of origin. The phylogenetic tree is rooted in Wuhan/WH01/2019 and Wuhan/Hu-1/2019 reference genomes.

and the vaccination campaign was progressing rapidly (Fig. 1a, b). After its initial detection, Delta spread rapidly across the island (Fig. 1b). Over 30% of the COVID-19 cases sampled and sequenced in June 2021 were caused by Delta. This variant has been characterized broadly as the most dominant emerging variant, replacing most lineages, and causing most of COVID-19 cases in the United States and Puerto Rico from its emergence through November 2021. To understand the rapid emergence and spread of Delta in the island, we reconstructed a maximum likelihood phylogenetic-focused tree with all Delta Puerto Rican sequences obtained in our dataset supplemented with additional sequences from Puerto Rico and the United States retrieved from GISAID with collection dates between May 1st, 2021, and September 30th, 2021. According to our phylogenetic inference, the emergence of Delta in Puerto Rico may have occurred between April 15th and June 14th, 2021, potentially after one or multiple introductions. The precise origin of the introductions was challenging to resolve, considering that multiple sequences from Mexico, the United States, and the Caribbean cluster among the early sampled sequences from Puerto Rico with low node support, <75% bootstrap value (Fig. 5). The first Delta lineage to be detected was B.1.617.2, which seems directly related to a small number of VBM B.1.617.1 (Kappa) that clustered basal to the focused tree. Tree topology is similar to the patterns observed in the Alpha focus tree, where more than 17 distinct clusters with sequences from Puerto Rico were observed diverging across

4 months of circulation in the island. Most of these clusters were associated with distinct Delta sub-lineages and seem closely related to similar sequences from the United States and the Caribbean. These clustering patterns and the diversity of Delta sub-lineages detected suggest that multiple introductions throughout 5–6 months propelled the emergence and transmission of this variant in the island.

## Discussion

Despite initial containment of COVID-19 cases in the spring of 2020, SARS-CoV-2 subsequently spread rapidly across the island propelled by multiple introductions and extensive intra-island transmission. Our partnership responded rapidly with the launch of this collaborative study aimed at enhancing the local capacity for genomic surveillance, understanding the development of the epidemic, and tracking viral lineages of public health concern. Our efforts also led to the early identification and tracking of most of the SARS-CoV-2 variants of lineages circulating in the island; information that was reported monthly to the Puerto Rico Department of Health to inform case investigation and epidemic response. As a result, our study contributed more than 75% of the genomes from the initial phase of the local epidemic through December 2020, and overall, to over 23% of the genomes sampled in Puerto Rico published in GISAID by October 31st, 2021.

Our systematic genomic surveillance and phylogenetic analyses suggest that the local epidemic was initiated and frequently boosted by travel-associated cases. This hypothesis is supported by the diversity of B.x lineages detected in the island during a period when the local population movement was restricted by stay-at-home orders and other response measures enforced by the local government[27,28]. In addition, the identification of multiple clusters of Puerto Rican sequences phylogenetically related to viruses found circulating primarily in the United States further supports this assessment. The increase in COVID-19 cases observed in the summer of 2020 coincided with lifting the stay-at-home order and increased traffic of tourists during a period of high SARS-CoV-2 transmission in the United States. Travel restrictions between countries with high incidence and the United States affected Puerto Rico by interrupting international travel but potentially increasing the volumes of domestic travel to and from the island. Consequently, we did not observe frequent clustering of Puerto Rican sequences with sequences from regions other than the United States. This is further supported by the difference in lineages circulating in the Americas and Puerto Rico[42]. The frequent mixing of viruses between Puerto Rico and the United States, the lineage diversity circulating in the island, and the low density of sampling at the location of origin generated phylogenetic uncertainty, limiting the capacity to resolve the precise origin of some local clusters and specific transmission chains[43]. The evolution and lineage turnover dynamics of SARS-CoV-2 in the greater Caribbean region remains uncertain considering the limitations of inter-island travel, direct flights to Puerto Rico, and genomic sampling during the study period.

Upon the emergence and rapid spread of SARS-CoV-2 during the first 10 months of the epidemic (March 2020–January 2021), we observed a shift in phylogenetic clustering patterns that most likely resulted from in situ viral evolution and adaptation to the local population or environment. This pattern was observed with the divergence of lineage B.1.588 from the persistent local transmission of B.1 lineage, possibly due to the founder effect. The fixation of two non-conservative mutations in the spike and nucleocapsid proteins seem to have improved fitness for local circulation. However, since this lineage was not considered a VBM, little is known about its phenotype or impact on other regions. Similar clustering patterns resembling adaptation, as well

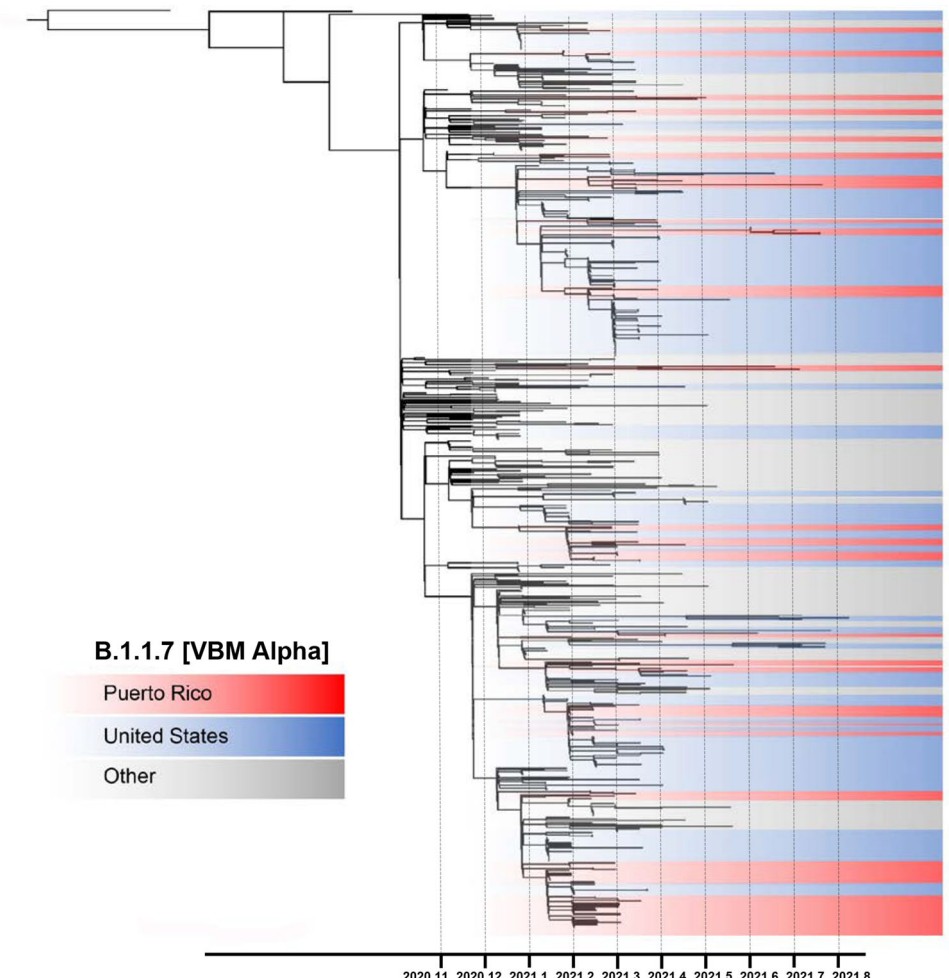

**Fig. 4 Emergence and spread of VBM Alpha in Puerto Rico driven by multiple introductions.** Phylogenetic reconstruction using a maximum likelihood tree inferred with 730 time-calibrated complete genomes, including 160 viral genomes from Puerto Rico and 570 contextual viral genomes from the United States and the Americas to provide a regional backdrop. Node structure supported by 1000 bootstrap replicates. Tree topology shaded in red represents clusters of viral genomes from Puerto Rico, blue shades represent clusters of genomes from the United States, and the gray shades represent clusters from other countries.

as the divergence of local lineages, have been observed in other regions of the world after a period of sustained local transmission leading to in situ evolution, including in the United Kingdom after the re-opening of a national lockdown[44–47]. Curiously, B.1.588 declined during a period of high transmission of SARS-CoV-2 and increased travel in which VBM Alpha was potentially introduced. These observations indicate the limited effectiveness of the initial efforts to prevent SARS-CoV-2 introductions and spread. The subsequent decline of the first epidemic wave could potentially be attributed to the combination of the local government response measures and the launch of the vaccination campaign.

The predominance of Alpha and the subsequent second epidemic wave could suggest that this variant presented a more virulent phenotype with a higher infectivity rate that outcompeted the autochthonous B.1.588 and other lineages in January 2021. This is consistent with the recurring observation that, once VBM/VOC emerged in the island, the frequency of other B.1x lineages was reduced substantially from that point forward. It is possible that this dynamic may have contributed to increased transmission during the initial phase of vaccination before the local population reached a 50% vaccination rate. Our findings are concordant with the epidemiological scenario in the United States[34,41,48].

As the vaccination campaign progressed during the spring of 2021, the number of cases decreased substantially, and the local government reduced most restrictions to public gatherings and indoor activities[27]. Puerto Rico then experienced another wave of increased travel during the summer of 2021 coinciding with the emergence of Delta. Though this variant was first detected in June 2021, our analyses estimated that the variant could have been introduced during the second epidemic wave in April 2021. If so, Delta could have faced a complex scenario with Alpha predominating circulation and almost half of the population vaccinated against the virus. However, the transmission of Delta displaced most of the circulating lineages and led to the third epidemic wave. Two different scenarios could be proposed for the emergence of Delta in Puerto Rico. First, Delta could have been introduced when Alpha dominated transmission, the two variants co-circulated, but Delta presented the phenotype that could outcompete Alpha. Key mutation patterns in Delta have been proposed to confer an advantage over other lineages[49]. Alternatively, the transmission of Alpha could have been already declining when Delta was introduced and emerged in a susceptible population. This second scenario proposed a gap between the dominance of Alpha and Delta. Recent reports from Madrid suggest that virus competition upon the emergence of Delta was not the exclusive factor driving the decline of Alpha but also a

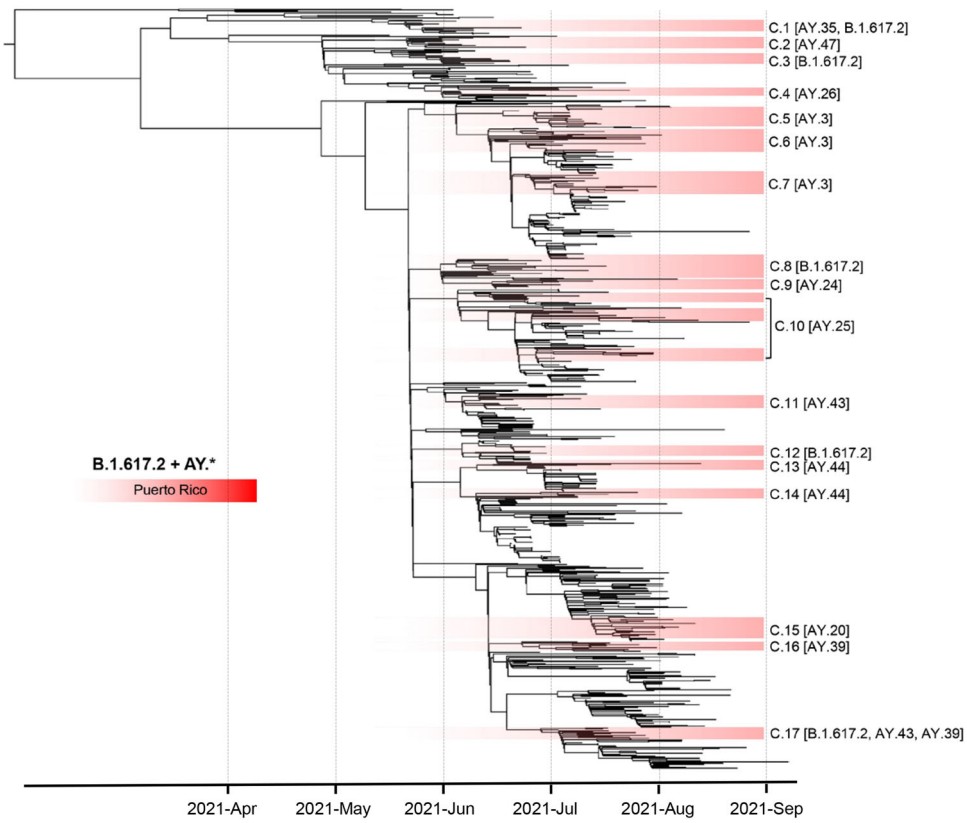

**Fig. 5 Emergence and spread of VOC Delta in Puerto Rico driven by the introduction of multiple sub-lineages.** Phylogenetic reconstruction using a maximum likelihood tree inferred with 815 time-calibrated complete genomes including 324 viral genomes from Puerto Rico and 491 contextual genomes from the United States and the Americas to provide a regional backdrop. Node structure supported by 1000 bootstrap replicates. Tree topology sections shaded in red represent clusters of viral genomes from Puerto Rico. Each cluster from Puerto Rico is labeled with cluster number C.x and Delta sub-lineage PANGO assignment [AY.x].

period of declined Alpha transmission facilitating the emergence of Delta in the region[50]. Though it is possible to speculate that Delta's phenotype exhibited some resistance to the vaccine, the decline of the third epidemic wave after ~4 months of Delta circulation, coincides with the population reaching a 77% vaccination rate with at least one dose.

The pattern of lineage turn-over observed in this study, where the predominant variants circulated for ~4 months and then were replaced by another variant in Puerto Rico, should be further compared with the transmission in larger countries with larger populations and human movement. Similar patterns have been observed in the United States, where variant emergence and spread potentially affect Puerto Rico[51]. Travel restrictions could have blocked the introduction of additional variants directly to Puerto Rico through international travel as seen in the difference in lineages detected between Puerto Rico and the rest of the Americas and Caribbean region[42]. However, our results suggest that lineage turn-over was in part driven by domestic travel in the United States. We also speculate that these cycles could be related to the limitations of an island's geography limiting access to the island only by restricted air travel at the time, the rapid response of the local government with restrictive measures to controls the spread of the virus and a vaccination campaign that reached over 84% of the population by December 1st, 2021[26].

Although our study sampled viral genomes from various cases and periods during the local pandemic, the universal ARTIC v3 tiled PCR-amplicon NGS workflow's sensitivity used in this study is limited to specimens with PCR Ct values below 28. Thus, we question if there are viral genetic differences from infections with lower viral loads or inconsistent variant-calling from samples

with lower viral loads that might affect the accuracy of lineage assignment[52–54]. The sampling for this study was also limited due to the ability of our partnership to procure samples from every municipality of the island, especially during the first year of the local epidemic. The availability of case metadata, such as travel history, was also limited, which would have facilitated an in-depth analysis on the impact of importations on the island. Future national genomic surveillance programs could benefit from improved systematic sampling engaging with clinical laboratories to ensure timely reporting of results to the local public health authorities, proper sample storage, and transfer to sequencing laboratories. Regarding phylogenetic analyses, the slow mutation rate and the low genetic diversity of the virus frequently impair the resolution of internal nodes with statistical support affecting the interpretation of phylogenetic histories and geographic origins. These analyses could also be affected by selecting context genomes from the unprecedented abundance of genomic data published in GISAID.

This study provides an overview of the COVID-19 epidemic in Puerto Rico during March 2020–November 2021 from the genomic epidemiology perspective. The documentation of an autochthonous lineage and dynamics of virus movement between the United States and Puerto Rico is important to inform prevention and surveillance efforts in both regions. Our phylogenetic study offers the genomic framework to understand the genomic changes occurring through time in the Puerto Rican viral population and elucidate the mutational landscape within this region. Furthermore, our ongoing genomic surveillance initiative will facilitate the study of SARS-CoV-2 intra-island phylodynamics and compare pre- and post-vaccination populations. Finally, this

report highlights the importance of government and academic partnerships to respond to public health threats and the potential of systematic genomic surveillance to improve disease prevention and control.

## Data availability

All genome sequences and associated metadata in this dataset are published in GISAID's EpiCoV database. To view the contributors of each individual sequence with details such as accession number, virus name, collection date, originating lab and submitting lab, and the list of authors, please visit https://doi.org/10.55876/gis8.220722zw[55] for contextual genomes and Supplementary Data 1 for the list of genomes generated by this study.

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

## Acknowledgements

We thank our partners from the Puerto Rico Department of Health, especially the staff from the Institute of Public Health Laboratories and the Biological and Chemical Emergencies Laboratory, for their contribution to the genomic surveillance framework and sample procurement. We also thank Dr. Anderson Brito and Chaney Kalinich, former members of the Dr. Nathan Grubaugh laboratory at the Yale School of Public Health, New Haven, CT, for providing invaluable technical assistance with data analysis. We acknowledge all healthcare workers and authors submitting data to GISAID, "Data availability" section, https://doi.org/10.55876/gis8.220722zw. The findings and conclusions in this report are those of the author(s) and do not necessarily represent the official position of the Centers for Disease Control and Prevention or the National Institute of Health. This project was partially funded by the Centers for Disease Control and Prevention's Advanced Molecular Detection Program and the COVID-19 Laboratory Task Force, the Puerto Rico Science, Technology and Research Trust (VRA), CDC U54MD007579 (VRA), and CDC U01CK000580 (VRA). Additional funding for this study was provided by the National Institute of General Medical Sciences of the National Institute of Health under award number U54GM133807 (NoA: 5U54GM133807-02) to Dr. Riccardo Papa. This project was also supported by UPR COVID-19 emergency funds #2020-2488 to Dr. Riccardo Papa.

## Author contributions

G.A.S. and J.L.M.J. conceived and designed the study with contributions from R.P., S.M.V.B., and E.O. B.F., G.L.G., K.N.C., L.C.H., and S.M.V.B. contributed to genomic sequencing, data analysis, and interpretation of results. G.A.S. drafted the manuscript with contributions from B.F., G.L.G., K.N.C., L.C.H., S.M.V.B., and J.L.M.J. H.R.V., V.R.A., L.E.A., M.M., L.H., I.C., E.O., G.P.B., R.P., and J.L.M.J. contributed to the revision and approved the final version of the manuscript.

## Competing interests

The authors declare no competing interests.
