## [Peer Review File · Communications Medicine]

Reviewers' comments:

Reviewer #1 (Remarks to the Author):

This manuscript is a nice epidemiologic piece and should be published. Some minor comments:

- 361. please include a reference for this statement.
- Consider the impact on viral circulation between Caribbean islands, variant distribution in the Dominican Republic are quite different with PR variant vs waves dynamics.

Reviewer #2 (Remarks to the Author):

Summary of work:

This work presents the results of a 19-month genomic surveillance study of SARS-CoV-2 carried out in Puerto Rico. Covid-19 cases were sampled monthly with 753 complete genome sequences generated by the study using Illumina sequencing. Phylogenetic results performed on the main lineages identified in Puerto Rico during the study period provided evidence for dates of importation but were unable to resolve with confidence possible locations of origins. Evidence is also presented for the emergence of an autochthonous B.1.588 lineage in Puerto Rico and the authors attempt to contextualize the changes in lineage frequency observed in Puerto Rico during the study period with various public health policies and mitigation efforts enacted in the country.

Overall Impression:

The work provides a thorough and comprehensive analysis of the main SARS-CoV-2 lineages circulating in Puerto Rico during the study period. It is well written, the data and results mostly clearly presented and worth publishing. However, the work presented provides an overall view of changing lineages in Puerto Rico and is not primarily focused on the emergence of the B.1.588 lineage as the title suggests. The Discussion can be improved by including comparisons to data from and studies conducted in other countries (other than the United States), specifically any island states.

Specific comments and suggestions:

Line 99 - The authors claim that Puerto Rico is an ideal setting in which to monitor and track SARS-CoV-2 lineage introduction and spread due to it being a geographically isolated location. This claim should be substantiated with further discussion and comparisons to similar studies conducted in other similar locations.

Line 130 - Mention is made of the data included in the analyses including these imported cases. However, no further mention is made of these cases. It would be worth indicating on the relevant figure where these sequences fall within the larger dataset and discussing in the text if it is known whether any of these cases contributed to introduction events to Puerto Rico.

Page 168, Figure C - Panel C can be moved to the Supplementary Information. It is referenced only once (page 7, lines 163-165) and the difference in the groupings of the dates used in panels A and B distract from the figure.

Line 177 - "... due to high frequency." Is this why it was considered separately or because it is a

lineage of focus due it possibly having emerged in Puerto Rico?

Line 189 - Is it possible to comment on the number of introductions given the results of the analysis, as well as comment on how much earlier that detection of the lineages by traditional epidemiological methods? This is worth discussing and for the main lineages presented and discussed in the paper.

Lines 220-222 - The replacement of B.1.588 and timeline stated is not clear in Figure 2 as expected given this text. Perhaps include a sub-panel showing the specific area of phylogeny being referred to? Or indicate using a graphic on the phylogeny the area?

Line 248 - Studies conducted in other countries and regions should be considered in order to better contextualize the results presented here.

Line 285 - Reference is made to "low node support". Support values over a certain threshold and those of interest should be indicated on the tree in Figure 5.

Line 347 - The use of the word "Interestingly" to describe the decline in frequency of B.1.588 observed is questionable given that Alpha was possible introduced at that point. An attempt is made to compare the fitness and infectivity rates of B.1.588 and Alpha in the next paragraph. However, given the focus on B.1.588 in this paper more discussion is needed on B.1.588 and its behaviour observed in other countries and comparisons of fitness, infectivity etc. to other main SARS-CoV-2 lineages.

Line 360-361 - Reference needed

Line 392 - More discussion is needed of the relevance of the island geography to the results presented (see comment for Line 99)

Reviewer #3 (Remarks to the Author):

This study describes the SARS-CoV-2 genomic epidemiology in Puerto Rico (PR) from the first detection in March 2020 through to September 2021. To do this, in association with the health agencies and academic institutions, the authors sequenced 753 genomes covering all health regions in PR. Key results include detection of multiple introductions, lineage turnover over time, and the emergence of B.1.588 locally, which subsequently spread to the U.S, and finally, thorough detailed analysis of the Alpha and Delta sequences the study show extensive migration between PR, the US and the Caribbean.

As Puerto Rico has experienced a severe pandemic, the data presented here has the potential to improve understanding of regional dynamics, however, the description of Results is not clear in many cases, and sometimes does not accurately portray the Results, and there was some confusion in epidemic terminology (Major comment below), indicating the manuscript needs a thorough revision for rigour and clarity. As the study remains largely descriptive, it is unclear how the epidemic or regional migration patterns changes as the control measures changed, except for the first wave.

Major.

1. Several problems with terminology make it difficult to follow. Mainly, “epidemic peaks/peaks” is incorrectly used throughout instead of “epidemic wave”. For example, in line 142 “Circulation of lineage B.1.588 declined during the **first peak**” of the epidemic in the winter of 2020” should instead be “first epidemic wave in the winter of”. “sub-tree” has a different meaning in phylogenetics - it should just be “tree” in most cases as these were constructed separately.

2. Statements in the Results that need further clarification

Results: Local epidemic and variant detection.

- Lines 118-119 “During March–July 2020, the number of confirmed COVID-19 cases remained low, associated with the strict stay-at-home order.” is not fully correct as substantial cases were detected in July as stated in the next sentence, indicating cases rose before lifting stay-at-home orders.
- Line 136-138 is not clear. “The initial phase of the epidemic was characterised by the detection of a wide diversity of B.1x lineages that circulated at low frequency for short periods of time, suggesting that the local epidemic was initiated by multiple introduction events.” Wide diversity is not apparent in Figure 1.
- Description in lines 148-150 does not match Figure 1B. The authors state “A steep reduction in confirmed cases was observed in the following months despite the predominant circulation of Alpha and the introduction of VBM 150 B.1.526 (Iota) in February 2021 and P.1/1.1 (Gamma) in April 2021 (Figure 1B).” However according to Figure 1B Alpha was not predominant until March 21.
- In-text citation to Figures/Tables can be improved. For instance, the description of wave one in PR is not clear.

Results: Phylogenetic reconstruction of the local pandemic

- Line 195-196 “Our analysis also showed the emergence and spread of the SARS-CoV-2 variants detected in Puerto Rico.” Rephrase as the spread in PR is not shown.
- Line 199-200 “The observed clustering patterns indicate multiple virus introductions with rapid and explosive expansion across the island in a short period of time.” The phylogenies indicate introductions, but the ‘explosive expansion’ is not apparent in Figure.

Results: Detection and spread of autochthonous lineage B.1.588

- This section is better explained, except for the part specifying the origins of B.1.588 from within Puerto Rico. The root of this tree is sparsely sampled with long branch lengths indicating a better sampling is needed in the tree Figure.

Results: Emergence of SARS-CoV-2 variants

- In this section, the authors describe the circulation of Alpha and Delta in more detail, showing extensive migration between PR and US, however, this section remains largely descriptive with the

same conclusion for both “multiple introductions throughout x-x months propelled the emergence and transmission of this variant in the island”. This could be combined together, and a summary of the number of migration events, or such quantities in relation to control measures over time could be a meaningful presentation of the introduction.

3. Methods

* Specify how the sequence alignment was treated. Specify if the sites deemed as problematic for phylogenetics have been removed.

* Yang96 was used for dating - specify why this codon-based model, and was the alignment trimmed to codon regions?

“concatenated” in line 707, should be changed to “combined”.

Delete “inference” in Line 711.

	Reviewer #1 comments	Authors' response and corrections
1	Line 361. please include a reference for this statement	The authors added references 41-43 on line 370 of the highlighted-marked version of the revised manuscript: Russell et al., 2022 reporting variant displacements in New York state, USA, Tordoff et al., 2021 reporting multiple importations into Washington state, USA and the continual seeding of the epidemic by a large number of introductions, mostly from within CONUS, and the CoVariants website developed by Emma Hodcroft which takes genomic data from GISAID and charts the lineage turnover by country including US states individually.
2	Consider the impact on viral circulation between Caribbean islands, variant distribution in the Dominican Republic are quite different with PR variant vs waves dynamics	The authors did not find published reports documenting lineage detection and evolution in the Caribbean islands; however, a study published by Leite et al. and the Pan American Health Organization report that the lineages circulating in the Americas and Caribbean region are different to the lineages circulating in Puerto Rico, which are more similar to the diversity in CONUS. This observation could be expected due to the international travel restrictions mandated by the US government, which covered Puerto Rico but allowed travel from CONUS to Puerto Rico. To further clarify this, the authors added "This is further supported by the difference in lineages circulating in the Americas and Puerto Rico" in lines 340-341 of the highlighted-marked version of the revised manuscript including the reference by Leite et al., 2022.
	Reviewer #2 comments	Authors' response and corrections
1	Line 99 - The authors claim that Puerto Rico is an ideal setting in which to monitor and track SARS-CoV-2 lineage introduction and spread due to it being a geographically isolated location. This claim should be substantiated with further discussion and comparisons to similar studies conducted in other similar locations.	During the first year of the epidemic in Puerto Rico, access to the island was limited to domestic air travel because the federal travel restrictions covered the jurisdiction of Puerto Rico. Therefore, importation of virus lineages originated mostly from CONUS. This is further evidenced by the diversity of lineages detected circulating in the Americas and the Caribbean regions which are different to the diversity circulating in Puerto Rico, Leite et al., 2022. To improve this discussion in the manuscript, we edited the following lines 400-403 of the highlighted-marked version of the revised manuscript "Travel restrictions could have blocked the introduction of additional variants directly to Puerto Rico through international travel as seen in the difference in lineages detected between Puerto Rico and the rest of the Americas and Caribbean region [Leite]... We also speculate that these cycles could be

		related to the limitations of an island geography limiting access to the island only by restricted air travel at the time..." lines 404-406.
2	Line 130 - Mention is made of the data included in the analyses including these imported cases. However, no further mention is made of these cases. It would be worth indicating on the relevant figure where these sequences fall within the larger dataset and discussing in the text if it is known whether any of these cases contributed to introduction events to Puerto Rico.	This study generated 753 complete genomes which are represented with red dots in the phylogeny in Figure 2. Because the initial imported cases were A lineage and no evidence of further spread, we considered not marking these in the phylogeny. However, we classified as imported cases those genomes obtained from patients with reported travel history or genomes closely associated to sequences from CONUS. Importation events were also inferred by detection of a wide diversity of B.1x lineages that predominated in CONUS, some of which we have no evidence of further spread in Puerto Rico considering the available sampling at the time. To clarify the limitation of travel history data, we added the following on lines 416-418 of the highlighted-marked version of the revised manuscript "The availability of case metadata, such as travel history, was also limited which would have facilitated an in-depth analysis on the impact of importations on the island."
3	Page 168, Figure C - Panel C can be moved to the Supplementary Information. It is referenced only once (page 7, lines 163-165) and the difference in the groupings of the dates used in panels A and B distract from the figure.	The authors prefer to maintain panel C as part of Figure 1 considering the evidence that provides supporting that multiple importation were received in the island harboring a variety of genotypes. Tick marks and columns representing 1 month are aligned between panels A and B to facilitate interpretation.
4	Line 177 - "... due to high frequency." Is this why it was considered separately or because it is a lineage of focus due it possibly having emerged in Puerto Rico	The authors considered B.1.588 as separate set due to both reasons listed by the reviewer. The frequency of B.1.588 genomes detected was substantially higher than other B.1x lineages at the time, in addition to the nature of the lineage having diverged in Puerto Rico as highlighted in this report. To clarify this, line 181 of the highlighted-marked version of the revised manuscript now reads "...due to high frequency and focus of this study."
5	Line 189 - Is it possible to comment on the number of introductions given the results of the analysis, as well as comment on how much earlier that detection of the lineages by traditional epidemiological methods? This is worth discussing	The authors cannot confirm with accuracy which of all the genomes samples come from imported cases considering the metadata available. Similar to comment #2, the authors classified as imported cases those genomes obtained from patients with reported travel history or genomes closely associated to sequences from CONUS. Importation events were also inferred by detection of a wide diversity of B.1x lineages, some of which we have no evidence of further spread considering the available sampling at

	and for the main lineages presented and discussed in the paper.	the time. Ancestor reconstruction allowed the inference of date of divergence or emergence of variants in the island. Since genomics continues to be the only methods for variant detection, cannot compare dates to detection by traditional epidemiological methods which rely on molecular diagnostics.
6	Lines 220-222 - The replacement of B.1.588 and timeline stated is not clear in Figure 2 as expected given this text. Perhaps include a sub-panel showing the specific area of phylogeny being referred to? Or indicate using a graphic on the phylogeny the area?	Figure 2 was cited incorrectly on lines 220-222. "Figure 2" was replaced with "Figure 1B" where it shows the timeline of the lineage replacement as described in the text, line 227 of the highlighted-marked version of the revised manuscript.
7	Line 248 - Studies conducted in other countries and regions should be considered in order to better contextualize the results presented here.	To further support our observation in context with reports from other regions, we added reference 41-43 in the Discussion section on line 370 of the highlighted-marked version of the revised manuscript "Our findings are concordant with the epidemiological scenario in the United States 41-43." These references present similar scenarios observed in the United States.
8	Line 285 - Reference is made to "low node support". Support values over a certain threshold and those of interest should be indicated on the tree in Figure 5.	The authors consider low node support those nodes with less than 75% bootstrap support. Considering the compressed graphical representation of the tree, the authors prefer not to include bootstrap values in the figure. To clarify node support, the authors added the following to line 294 of the highlighted-marked version of the revised manuscript "...with low node support, less than 75% bootstrap value (Figure 5)."
9	Line 347 - The use of the word "Interestingly" to describe the decline in frequency of B.1.588 observed is questionable given that Alpha was possible introduced at that point. An attempt is made to compare the fitness and infectivity rates of B.1.588 and Alpha in the next paragraph. However, given the focus on B.1.588 in this paper more discussion is needed on B.1.588 and its behaviour observed in other countries and comparisons of	The authors replaced the word "Interestingly" for "Curiously" because we find "curious" the switch of predominant lineage during a high transmission period suggesting potential competition of lineages in population whose immunological scenario was changing due to the vaccination campaign. However, the authors are not able to provide additional discussion on the behavior of B.1.588 because little is known about this lineage and it has not been reported by other groups, to date. To clarify this, we added the following statement to lines 353-354 of the highlighted-marked version of the revised manuscript "However, since this lineage was not considered a VBM, little is known about its phenotype or impact on other regions."

	fitness, infectivity etc. to other main SARS-CoV-2 lineages.	
10	Line 360-361 - Reference needed	To further support our observation in context with reports from other regions, we added reference 41-43 in the Discussion section on line 370 of the highlighted-marked version of the revised manuscript: Russell et al., 2022 reporting variant displacements in New York state, USA, Tordoff et al., 2021 reporting multiple importations into Washington state, USA and the continual seeding of the epidemic by a large number of introductions, mostly from within CONUS, and the CoVariants website developed by Emma Hodcroft which takes genomic data from GISAID and charts the lineage turnover by country including US states individually.
11	Line 392 - More discussion is needed of the relevance of the island geography to the results presented (see comment for Line 99)	Lines 400-406 of the Discussion section of the highlighted-marked version of the revised manuscript were edited as such: “Travel restrictions could have blocked the introduction of additional variants directly to Puerto Rico through international travel as seen in the difference in lineages detected between Puerto Rico and the rest of the Americas and Caribbean region (Leite et al, 2022). However, our results suggest that lineage turnover was in part driven by domestic travel with the United States. We also speculate that these cycles could be related to the limitations of an island geography limiting access to the island only by restricted air travel at the time...”.
	Reviewer #3 comments	Authors’ response and corrections
1	Several problems with terminology make it difficult to follow. Mainly, “epidemic peaks/peaks” is incorrectly used throughout instead of “epidemic wave”. For example, in line 142 “Circulation of lineage B.1.588 declined during the **first peak** of the epidemic in the winter of 2020“ should instead be “first epidemic wave in the winter of”. “sub-tree” has a different meaning in phylogenetics - it should just be “tree” in most cases as these were constructed separately.	The authors agree. The correct term should be epidemic wave. We have replaced the word “peak” with “wave” throughout the manuscript. The term “sub-tree” was used to refer to a focused tree on a smaller subset of the parental dataset. To clarify this, we replaced the term “sub-tree” with “focused phylogenetic tree” throughout the manuscript as well.

2	Lines 118-119 “During March–July 2020, the number of confirmed COVID-19 cases remained low, associated with the strict stay-at-home order.” is not fully correct as substantial cases were detected in July as stated in the next sentence, indicating cases rose before lifting stay-at-home orders.	The authors confirm that the steep increase in cases followed the lifting of the stay-at-home order issued by the local government, through the exact timing of the steep increase was difficult to assess considering the accessibility and precision of the data reported by the local government surveillance portal. To accommodate this uncertainty, we now start the statement in line 119 of the highlighted-marked version of the revised manuscript with “Around the time when the order was lifted...” The next sentence states that the following epidemic wave was detected in November 2020, line 122.
3	Line 136-138 is not clear. “The initial phase of the epidemic was characterised by the detection of a wide diversity of B.1x lineages that circulated at low frequency for short periods of time, suggesting that the local epidemic was initiated by multiple introduction events.” Wide diversity is not apparent in Figure 1.	The authors claim that a wide diversity of B.1x lineages were detected during the first year of the local epidemic. All of these B.1x lineages are grouped within the “Other” category, lavender colored bar in Figure 1B. The exact number of B.1x lineages detected during this period changed through time considering the changes in lineage definitions employed by Pangolin lineage assignment tool. More than 63 lineages. The legend of Figure 1, line 180 of the highlighted-marked version of the revised manuscript no clarifies “Non-VBM/VOC lineages (n > 63) are categorized as a collective labeled “Other”.
4	Description in lines 148-150 does not match Figure 1B. The authors state “A steep reduction in confirmed cases was observed in the following months despite the predominant circulation of Alpha and the introduction of VBM 150 B.1.526 (Iota) in February 2021 and P.1/1.1 (Gamma) in April 2021 (Figure 1B).” However according to Figure 1B Alpha was not predominant until March 21.	To clarify the statement, lines 149-153 of the highlighted-marked version of the revised manuscript now read “A steep reduction in confirmed cases was observed in the following months despite the introduction of VBMs B.1.526 (Iota) and P.1/1.1 (Gamma) in February and March 2021 respectively and the predominant circulation of Alpha in March 2021(Figure 1B).”
5	In-text citation to Figures/Tables can be improved. For instance, the description of wave one in PR is not clear.	Lines 121-122 of the highlighted-marked version of the revised manuscript reference Figure 1A and indicate that the peaks of each epidemic waves occurred in November 2020, April 2021, and August 2021. To clarify in-text citation, line 121 now reads “...we observed 3 epidemic waves with high points in November 2020, April 2021, and August 2021.” The reader can now refer to each epidemic wave in Figure 1A.
6	Line 195-196 “Our analysis also showed the emergence and spread	The authors claim that detection of multiple local cases of each variant reflect spread of the variant in the island. To clarify the

	of the SARS-CoV-2 variants detected in Puerto Rico.” Rephrase as the spread in PR is not shown.	statement in lines 195-196, the authors replaced the term “spread” with “evolution” presented as phylogenetic data.
7	Line 199-200 “The observed clustering patterns indicate multiple virus introductions with rapid and explosive expansion across the island in a short period of time.” The phylogenies indicate introductions, but the ‘explosive expansion’ is not apparent in Figure.	Similar to comment #6, the authors claim that ‘explosive expansion’ refers to the multiple cases recorded rapidly after the introduction of a variant was detected. Evidence of this rapid expansion is presented in Figure 1B through the changes in frequency of detection through time and the variable branch lengths in the global phylogenetic tree in Figure 2. To clarify the statement in lines 203-205 of the highlighted-marked version of the revised manuscript, the sentence now reads “The observed clustering patterns in the phylogenetic trees and the rapid increase in frequency following initial detection indicate multiple virus introductions with swift expansion across the island...”
8	This section is better explained, except for the part specifying the origins of B.1.588 from within Puerto Rico. The root of this tree is sparsely sampled with long branch lengths indicating a better sampling is needed in the tree Figure.	The authors agree; however, the phylogenetic tree was reconstructed with the genome sequences sampled during the time. Additional sampling would be needed to clarify the precise origin of B.1.588 but those genomes are not available at this time. Lines 353-354 and 416-418 of the Discussion section clarify the limitations with sampling and metadata availability.
9	In this section, the authors describe the circulation of Alpha and Delta in more detail, showing extensive migration between PR and US, however, this section remains largely descriptive with the same conclusion for both “multiple introductions throughout x-x months propelled the emergence and transmission of this variant in the island”. This could be combined together, and a summary of the number of migration events, or such quantities in relation to control measures over time could be a meaningful presentation of the introduction.	The authors agree with this reviewer over the fact that both Alpha and Delta sections present similar findings though with different variants over different periods of time. However, the authors prefer to maintain the sections separated since the multiple introductions of Delta is also supported by the multitude of sub-variants introduced in the island, indirectly suggesting that Delta circulated over a longer period of time with a higher genomic diversity. In addition, the authors are currently working on an additional manuscript reporting the decline of Delta and replacement by Omicron; a report that would benefit from the data provided here in the Delta section.
10	Specify how the sequence alignment was treated. Specify if the sites	Sequence alignment was performed with MAFFT independently or within the Nextstrain/ncov workflow with the same datasets throughout the study as indicated in line 695. The workflow is

	deemed as problematic for phylogenetics have been removed	designed to filter sequences with lower quality, align the sequences and verify alignment quality. No additional revision was performed to the dataset after running the Nextstrain/ncov workflow.
11	Yang96 was used for dating - specify why this codon-based model, and was the alignment trimmed to codon regions?	Yang96 was the best fitting model which produced data with sufficient convergence (ESS>200). Yes, the alignment was trimmed to coding sequence. To clarify this, we added “trimmed to coding sequence” in line 714 of the highlighted-marked version of the revised manuscript.
12	“concatenated” in line 707, should be changed to “combined”	The word “concatenated” was replaced by the word “combined” in line 701 of the highlighted-marked version of the revised manuscript.
13	Delete “inference” in Line 711	The word “inference” was deleted in line 705 of the highlighted-marked version of the revised manuscript.

REVIEWERS' COMMENTS:

Reviewer #1 (Remarks to the Author):

Requested revisions were fully addressed, fantastic work!

Reviewer #2 (Remarks to the Author):

The authors have satisfactorily addressed the majority of the comments. My remaining concern is, while I understand there are few studies published at this time, the discussion of SARS-CoV-2 lineage detection and evolution in the wider Caribbean region is limited. A more robust discussion would help to better contextualize the results presented.

The overall study, methodology and results and conclusions presented here are not novel. However, the methodology is sound and the results obtained would be of interest to other researchers working on SARS-CoV-2 and viral transmission dynamics within the Caribbean region.

Reviewer #3 (Remarks to the Author):

The authors have addressed all of my comments and made adequate changes to the revised manuscript. I have no further comments. Congratulations to the authors.

	Reviewer #2 comments	Authors' response and corrections
1	The authors have satisfactorily addressed the majority of the comments. My remaining concern is, while I understand there are few studies published at this time, the discussion of SARS-CoCV-2 lineage detection and evolution in the wider Caribbean region is limited. A more robust discussion would help to better contextualize the results presented. The overall study, methodology and results and conclusions presented here are not novel. However, the methodology is sound and the results obtained would be of interest to other researchers working on SARS-CoV-2 and viral transmission dynamics within the Caribbean region.	The authors recognize the need to understand the evolution and lineage turn-over dynamics in the wider Caribbean region, especially in context with Puerto Rico. Unfortunately, this is difficult to assess at this point considering that the sampling of SARS-CoV-2 genomes representative of the wider Caribbean region was limited during the study period. In addition, travel between the islands is usually limited with scarce direct flights to/from Puerto Rico, Travel restrictions imposed by the US government limited inter-island travel further by interrupting cruise ship travel. The authors have included the following statement in the Discussion section, lines 287-289, "The evolution and lineage turnover dynamics of SARS-CoV-2 in the greater Caribbean region remains uncertain considering the limitations of inter-island travel, direct flights to Puerto Rico, and genomic sampling during the study period."